# Neck Schwannoma Masking as Thyroid Tumour: Into the Deep of Diagnostics and Anatomy

**DOI:** 10.3390/diagnostics14202332

**Published:** 2024-10-19

**Authors:** Serghei Covantsev, Anna Bumbu, Anna Sukhotko, Evghenii Zakurdaev, Ivan Kuts, Andrey Evsikov

**Affiliations:** 1Department of Clinical Research and Development, Botkin Hospital, 125284 Moscow, Russia; 2Department of Oncology, Botkin Hospital, 125284 Moscow, Russia; 3Department of Pathology, Botkin Hospital, 125284 Moscow, Russia

**Keywords:** thyroid, schwannoma, fine-needle aspiration, core-needle biopsy, tumours, nervous system

## Abstract

Schwannomas are benign nerve sheath tumours that exhibit a slow rate of growth. In the vast majority of cases, schwannomas manifest as asymptomatic masses. The presence of symptomatic lesions may necessitate surgical removal. The incidence of schwannomas ranges from 4.4 to 5.23 cases per 100,000 population, accounting for approximately 7% of all primary tumours in the central nervous system. There is a limited number of case reports describing schwannomas outside the central nervous system. In rare instances, schwannomas may originate at the level of the thyroid gland. In such cases, incidental neck schwannomas may be mistaken for thyroid or parathyroid tumours. The increasing incidence of thyroid cancer draws more attention to all thyroid nodules, both benign and malignant. Thyroid nodules are detected in up to 65% of autopsies, with only 4–6.5% being malignant. Thyroid tumours are typically diagnosed by USG; however, they are often revealed incidentally during neck CT or MRI for other conditions. To rule out malignancy, tumour verification is required. The modern diagnosis of thyroid cancer is based on fine-needle aspiration (FNA) biopsy and cytology, which is classified according to the Bethesda classification system. However, not all FNAs are informative, and the differential diagnosis and treatment strategies in cases of unsatisfactory results are not standardized, leading to potential intraoperative challenges. We present a case study of a patient with a thyroid nodule that was ultimately diagnosed with a schwannoma of the neck according to core-needle biopsy.

## 1. Introduction

Schwannomas are benign nerve sheath tumours that exhibit a slow rate of growth. In the vast majority of cases, schwannomas manifest as asymptomatic masses. The presence of symptomatic lesions may necessitate surgical removal [1]. The incidence of schwannomas ranges from 4.4 to 5.23 cases per 100,000 population, accounting for approximately 7% of all primary tumours in the central nervous system [2,3]. There is a limited number of case reports describing schwannomas outside the central nervous system. They usually appear during the 5th-6th decade with no sex predisposition. Approximately 20–50% of schwannomas arise in the head and neck region; however, schwannoma may be located anywhere in the body. Of these, 10–30% originate from Schwann cells of the vagus nerve, while another 10–20% arise from the sympathetic nervous system [4,5,6]. Despite the fact that the head and neck are one of the primary sites of occurrence, only 250 patients have been reported in the literature [7]. These cases primarily involve schwannomas affecting cranial nerves (such as V, VII, IX, X, XI, and XII), as well as sympathetic or peripheral nerves [6,8]. High-volume surgical centres typically see from 30 to 50 such cases every 8–9 years [5,6]. In rare instances, schwannomas may originate at the level of the thyroid gland. In such cases, incidental neck schwannomas may be mistaken for thyroid or parathyroid tumours due to their anatomical closeness to these structures [9,10,11,12,13,14,15].

Thyroid cancer incidence rates have increased in many countries and settings. Globally, in 2020, the incidence of thyroid cancer was 10.1 per 100,000 women and 3.1 per 100,000 men [16]. This increase in the number of thyroid cancer cases resulted in close attention being paid to all thyroid nodules, both benign and malignant. Thyroid nodules are detected in up to 65% of autopsies, with only 4–6.5% being cancerous [15,16,17,18]. Thyroid tumours are typically diagnosed by USG; however, they are often revealed incidentally during neck CT or MRI for other conditions. To rule out malignancy, tumour verification is required. The modern diagnosis of thyroid cancer relies on fine-needle aspiration biopsy (FNA) and cytology, which is classified according to the Bethesda classification system [19].

However, not all FNAs are informative, and the differential diagnosis and treatment strategies in cases of unsatisfactory results are not standardized, leading to potential intraoperative challenges. We present a case study of a patient with a thyroid nodule that was ultimately diagnosed with a schwannoma of the neck according to core-needle biopsy (CNB).

## 2. Case Description

During an annual medical examination in 2022, a patient, a 32-year-old male, was found to have a proliferative mass in the neck area. Ultrasound examination of the thyroid gland revealed an oval hypoechoic heterogeneous formation with a size of 18 × 21 × 31 mm, clear, well-defined boundaries, and areas of abnormal blood flow in the projection of the left inferior parathyroid gland along the posterior surface of the left lobe of the thyroid gland (see Figure 1). Magnetic resonance imaging showed a volumetric formation with clear boundaries and heterogeneous contents measuring 33 × 24 × 25 mm on the lateral posterior surface of the left lobe of the thyroid gland, which looks hyperintensive on the T2-weighted image and has no signs of limited diffusion (see Figure 2).

The computed tomography scan also revealed a mass with a size of 21 × 24 × 25 mm with a density of 18 Hounsfield units that accumulated contrast agent and was situated between the thyroid gland and the vertebral column (Figure 3). The parathyroid hormone level was 4.06 pmol/L, and the ionized calcium concentration was 1.22 mmol/L. Thyroid-stimulating hormone T3 and T4 were also within the normal range and comprised 2.7 mIU/L, 1.7 nmol/L and 89 nmol/L concurrently. Therefore, the patient had normal thyroid and parathyroid function. Also, there were no other laboratory deviations recorded at the time of hospitalization. The patient did not have any previous medical conditions and considered himself healthy.

The results of FNA were unsatisfactory and classified as Bethesda category I. A subsequent FNA was also non-diagnostic. To make a decision regarding diagnostic hemithyroidectomy, the patient was referred to an endocrine surgeon. Due to uncertainty about the diagnosis, the patient underwent CNB of the cervical region mass upon admission. Histopathological analysis revealed a schwannoma (Figure 4). Biopsy specimens contained spindle-shaped cells with areas of myxoid structure. Apart from that, characteristic palisade patterns of nuclei (Verocay bodies) and rare mitoses were noted (WHO grade I tumours).

The surgical procedure was conducted in aseptic conditions, under general anesthesia. Following meticulous marking, a small incision in the form of a collar was made on the left side of the neck, in the vicinity of the mass. The skin, subcutaneous fat, and platysma were accurately dissected, allowing for the identification of the sternocleidomastoid muscle and the common carotid artery, along with an adjacent volumetric mass measuring approximately 3 cm in diameter (see Figure 5).

The apical margin of this formation was positioned at the level of the left lobe of the thyroid gland, while its inferior border approached the jugular notch. Laterally, the mass extended towards the left carotid artery, while its distal margin came into contact with the body of the lumbar vertebrae. Utilizing a dissector and cautery pencil, the mass was meticulously excised. A flexible drainage tube was inserted into the tumour bed, followed by the creation of a contraperture to facilitate the removal of any exudate. Once haemostasis was achieved in the surgical area, skin sutures were applied, and a sterile dressing was secured over the wound. The catheter was removed one day after the surgical procedure, and the patient was discharged two days later without any complaints.

The postoperative histological examination confirmed the findings of the CNB. Immunohistochemical analysis revealed that the tumour exhibited a positive reaction to S100 (100%) and a negative response to CD34 and SMA markers. The Ki67 index was 2%.

Following a two-year follow-up period, no signs of tumour recurrence or neurological deficits were observed.

## 3. Discussion

In up to 90% of cases, schwannomas are usually solitary and sporadic; however, they can be associated with type 2 neurofibromatosis or other genetic conditions [20]. Schwannomas of the cervical region stem from the vagus nerve in approximately 11–20% of cases. They can also originate from the sympathetic chain, which comprises 11–34% of cases, or the cervical plexus and the brachial plexus, which account for 3–50% and 17% of cases, respectively. In rare instances, schwannomas may result from the hypoglossal nerve (3–6%). Moreover, in some cases they can originate from the greater auricular nerve (5.6%), the hypoglossal nerve (3–5.55%), or even the recurrent laryngeal nerve (3%). The latter can lead to a misdiagnosis, as these tumours can be closely adjacent to the thyroid gland. In up to 17% of cases the exact origin of the nerve cannot be determined [5,6]. Figure 6 demonstrates the nerves, possible origins of schwannoma, and other neoplasms, which are located in close proximity to the thyroid. Tumours arising from some of these nerves can only be misidentified when they are large, with one of the tumour poles situated close to the thyroid gland. Schwannomas must be differentiated from paragangliomas and other nerve-associated tumours, such as neurofibromas, granular cell myoblastomas, neurogenic sarcomas, and melanomas [21].

The first case report of a primary schwannoma of the thyroid gland was published in 1964 by Delaney and Fry [22]. Approximately 60% of tumours in the neck region present as asymptomatic, palpable masses, and 10% are incidentally discovered in imaging studies. Schwannomas are usually asymptomatic and can show no signs of disease for more than 10 years. On average, the growth rate of these lesions does not exceed 1–3 mm each year [1]. In 20% of cases, patients present with neurological deficits, and in 10% of cases, pain and obstruction of the neck veins are the prevalent symptoms. There have been a limited number of reports regarding thyroid schwannomas (Table 1). The vast majority of patients are 30–35-year-old females, who are typically concerned with a neck mass. Reports of male patients affected are less common. Progressive swelling in the neck region and Horner’s syndrome may indicate malignancy but can also occur in benign tumours [21,23]. Furthermore, as the neoplasm expands, surgical excision of the tumour may give rise to a variety of complications, including the so-called first-bite syndrome [1]. First-bite syndrome (FBS) is a rare complication that occurs after damage to the sympathetic chain’s superior cervical ganglion (SCG). With destruction of the SCG, the patient can develop FBS and Horner syndrome. This condition can be seen in patients who undergo parapharyngeal space surgery and can result in the development of severe parotid gland pain at the first bite of food [24].

In non-contrast computed tomography (CT) scans, schwannomas often exhibit a decreased density compared to muscle tissue, appearing hypodense. Schwannomas consist of spindle cells that have two growth patterns: Antoni type A and Antoni type B. In contrast-enhanced CT images, schwannomas predominantly exhibiting the Antoni A pattern manifest as solid, intensely enhancing, heterogeneous, and hypodense masses. Schwannomas characterized by the predominance of Antoni B typically present as pseudocysts, with minimal or no enhancement in contrast-enhanced images. On MRI, these lesions exhibit varying levels of intensity, ranging from low to equal intensity of normal tissue on T1-weighted sequences and high intensity on T2-weighted images, depending on their cellular composition. Contrast-enhanced T1-weighted MRI reveals moderate to significant enhancement of these lesions. The appearance of these lesions is generally homogeneous for smaller ones and becomes heterogeneous as they increase in size. In certain occasions, the pronounced hyperintense signal observed on T2-weighted MRI scans serves as a means of distinguishing the mass from the adjacent, less hyperintense structures such as the thyroid gland. The existence of a discernible boundary between the mass and the thyroid gland is indicative of the presence of an independent neoplasm [25].

The FNA biopsy of a tumour located near the thyroid gland is often inconclusive, as in most cases, the cytology reports fall into categories I, II, III, or IV according to the Bethesda system [9,10,12,13]. The specificity of FNA and imaging studies for diagnosing schwannomas is approximately 20% and 38%, respectively [6]. Table 1 presents a comparison of FNA results, indicating that in the prevalent number of cases, they are inconclusive.

During cytological evaluation, these tumours can be mistaken for other spindle cell lesions of the thyroid, such as smooth muscle tumours and spindle cell lesions [23]. The differential diagnosis of spindle cell tumours is a broad field that encompasses a diverse range of entities. These include neural-derived tumours, such as schwannomas, as well as mesenchymal neoplasms like leiomyomas and solitary fibrous tumours. Hemangiopericytomas are also included in this category. Epithelial tumours, such as anaplastic thyroid carcinomas, medullary thyroid carcinomas, thymomas, and spindle epithelial tumours with thymus-like features (SETTLE), are part of the spectrum as well. Moreover, hyalinizing trabecular adenomas deserve consideration. Immunohistochemical staining for S-100 protein has proven to be highly valuable for preoperative diagnosis of these neoplasms, especially in the case of schwannomas [10].

It is worth noting that non-thyroid lesions can sometimes be misdiagnosed as primary thyroid neoplasms if the FNA biopsy includes thyroid tissue, due to the trajectory of the needle through the gland prior to reaching the target lesion [26]. Consequently, the preoperative diagnosis is seldom established based on these investigations. Nonetheless, some experts advocate for CNB, as it can detect schwannomas more accurately.

**Table 1 diagnostics-14-02332-t001:** Thyroid schwannoma cases.

Age, Sex	Cytology	Clinical Picture	Reference
87, female	Bethesda III	Radiating pain after aspiration	[9]
70, male	Bethesda IV	Palpable mass, hoarseness, neck discomfort	[10]
30, female	Bethesda I	Swelling in the neck	[12]
47, male	Bethesda II	Progressive swelling in the neck	[13]
31, male	Schwannoma	Growing nodule in the neck	[3]
60, female	Schwannoma	Dysphagia and a significant increase in goitre size	[25]
33, female	Bethesda I	Palpable neck mass	[27]
33, female	Bethesda I	Large palpable mass, compression	[28]
26, female	Bethesda I	Neck mass	[29]
35, female	Bethesda I	Neck mass, Horner’s syndrome	[21]
63, female	Bethesda I	Foreign body sensation with swallowing	[30]
23, female	Paucicellular sample with occasional follicular cells of equivocal diagnostic value	Neck mass	[31]
32, male	Bethesda I	Neck mass	Current case

The imaging of neck tumours that may be associated with thyroid cancer can involve various techniques, such as ultrasonography (USG), CT with contrast enhancement, MRI with contrast enhancement, and positron emission tomography (PET) or scintigraphy. A comparative analysis of these imaging modalities can be found in Table 2.

There are several publications that suggest CNB of the thyroid gland is a safe procedure that can provide valuable additional information in cases when cytology results are inconclusive [32,33]. This is particularly relevant in instances of rare thyroid gland tumours or tumours that mimic thyroid tumours. Specific diagnoses, such as “suspicious schwannoma” and “consistent with schwannoma”, can be made in up to 96.6% of CNB samples, while FNA yields such diagnoses in only 19.2% of cases (*p* < 0.001) [34].

Immunohistochemistry is crucial in this context, as the tumour should be positive for S-100 protein and negative for calcitonin, carcinoembryonic antigen (CEA), thyroglobulin, thyroid transcription factor 1 (TTF1), melan-A, and melanoma-associated antigen (HMB45), which helps to rule out schwannomas, medullary thyroid cancers, and well-differentiated thyroid cancers [25].

However, CNB remains somewhat controversial due to the potential risks of neuropathic pain or neurological deficits resulting from axonal damage [35]. Some experts recommend intracapsular resection of schwannomas with electrical nerve monitoring to reduce the risk of nerve injury [36].

A complete excision is strongly advised in order to prevent both the progression and recurrence of schwannomas [1,7]. These are slow-growing tumours that have a propensity to displace adjacent fascicles. Failure to excise them can result in their continued growth, leading to a myriad of clinical manifestations, including nerve dysfunction and paresthesia.

However, in certain cases, it is hard to distinguish between an extra-thyroidal mass and an intrathyroidal lesion, leading to unnecessary thyroid surgery. Frozen section pathology during surgery might be helpful in such situations to guide the surgeon [10]. Partial resection carries a high risk of recurrence (greater than 50%), as well as the possibility of malignant transformation (4%). A complete excision is strongly advised to prevent both progression and recurrence. Additionally, an increase in size can lead to potential hemorrhage, cystic degeneration, and necrosis, as reported in the literature [1,7,37].

Therefore, in this particular case of thyroid neoplasia, the patient had undergone several FNAs with no diagnostic result and was referred for hemithyroidectomy. CNB allowed us to obtain preoperative diagnosis and, since the tumour was not within the thyroid gland, abstain from hemithyroidectomy. The multidisciplinary approach that involved a medical ultrasonographer, radiologist, pathologist, and surgeon enabled us to obtain the diagnosis and choose adequate treatment tactics.

## 4. Conclusions

Schwannoma of the thyroid is a rare pathological condition, with only a handful of cases reported in the scientific literature. FNA is the standard method for diagnosing thyroid malignancies in the presence of suspicious nodules. However, its accuracy in the context of rare tumours is often limited. In such cases, CNB often provides more reliable results, albeit with a higher risk associated with the procedure. The benefit of a CNB is that the obtained histological specimen provides a tissue sample, unlike FNA, and the specimen can be studied by means of immunohistochemistry. The decision to proceed with CNB must be carefully balanced against the importance of obtaining a definitive diagnosis prior to surgery.

## Figures and Tables

**Figure 1 diagnostics-14-02332-f001:**
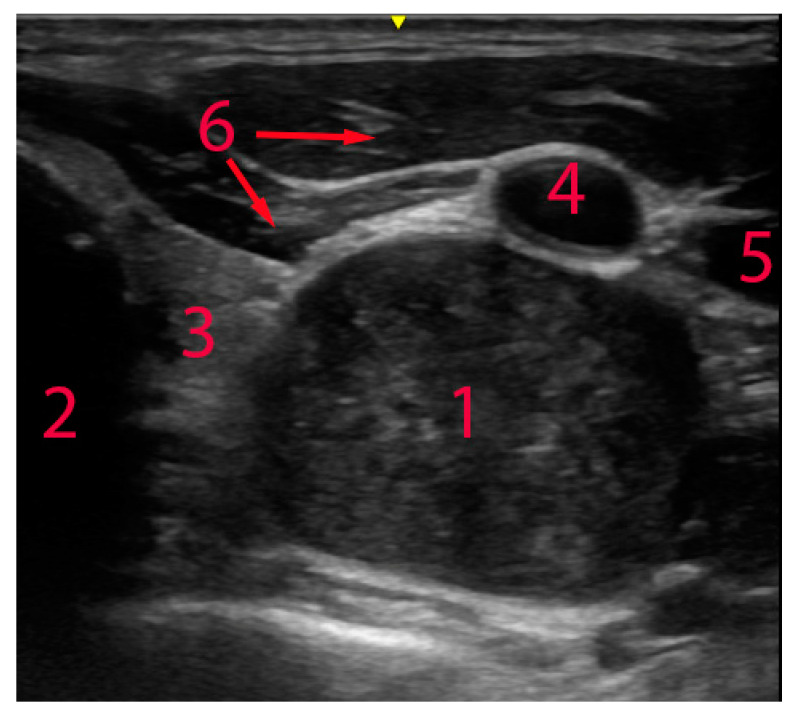
USG of the neck region at the level of the thyroid gland. 1—tumour, 2—trachea, 3—thyroid, 4—carotid artery, 5—jugular vein, 6—pretracheal muscles.

**Figure 2 diagnostics-14-02332-f002:**
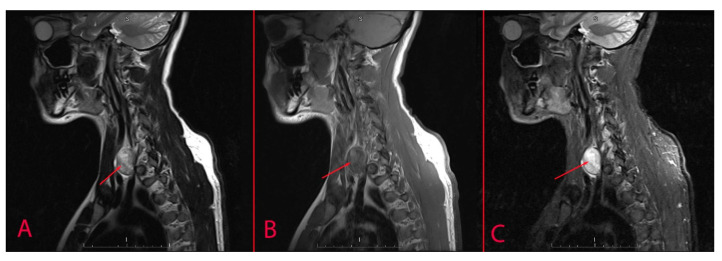
Soft tissue MRI (arrow indicates the tumour). (**A**)—T2, (**B**)—T1, (**C**)—T2 TIRM.

**Figure 3 diagnostics-14-02332-f003:**
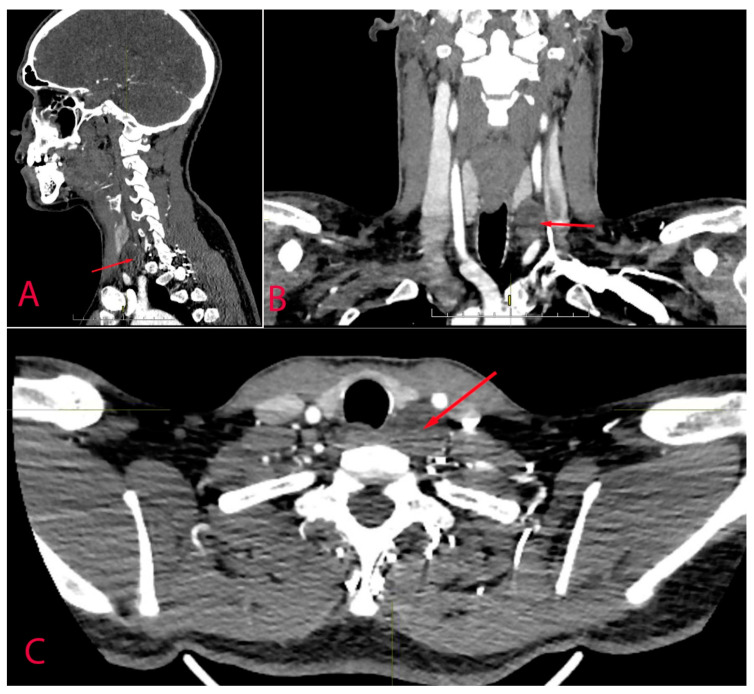
Contrast-enhanced CT scan of the neck at the level of the thyroid gland (arrow indicates the tumour). (**A**)—sagittal plane, (**B**)—frontal plane, (**C**)—axial plane.

**Figure 4 diagnostics-14-02332-f004:**
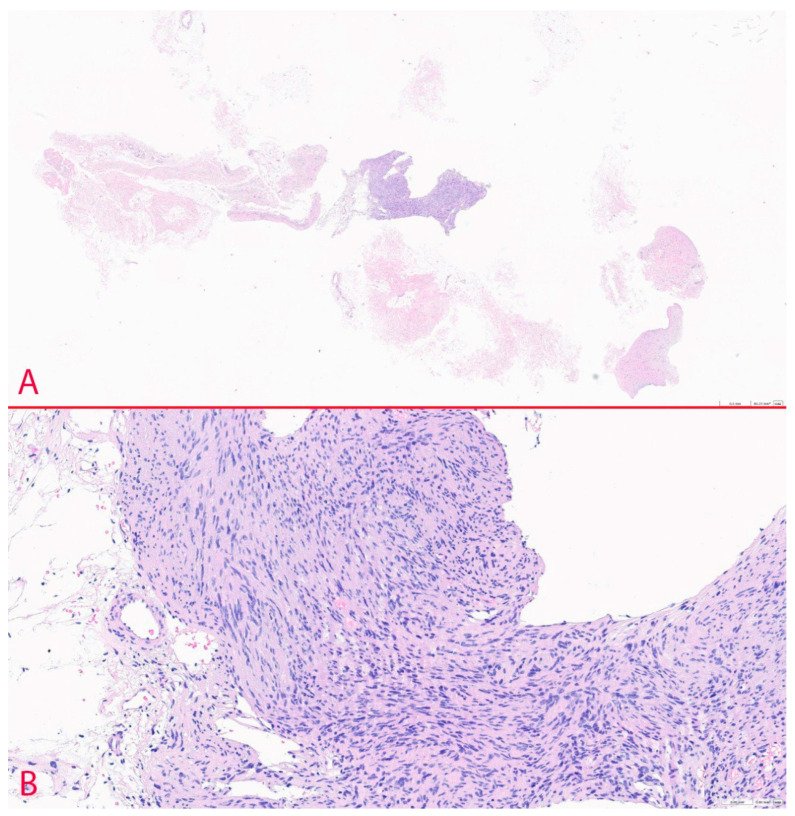
Histology of the specimens. (**A**)—CNB specimens (×5; H&E staining), (**B**)—CNB histology (×20 H&E staining).

**Figure 5 diagnostics-14-02332-f005:**
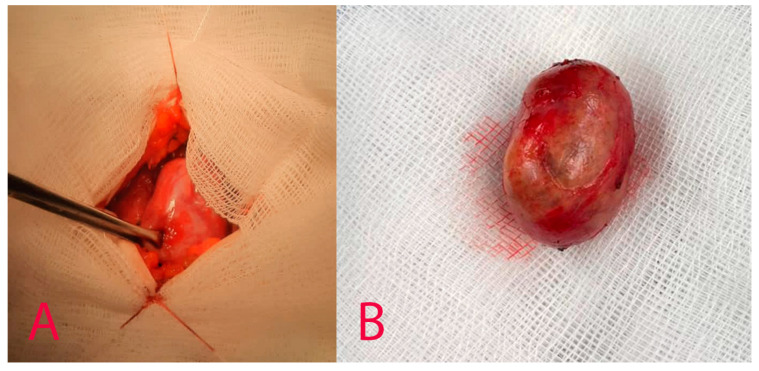
Macroscopic tumour appearance. (**A**)—intraoperative image of the tumour, (**B**)—postoperative specimen.

**Figure 6 diagnostics-14-02332-f006:**
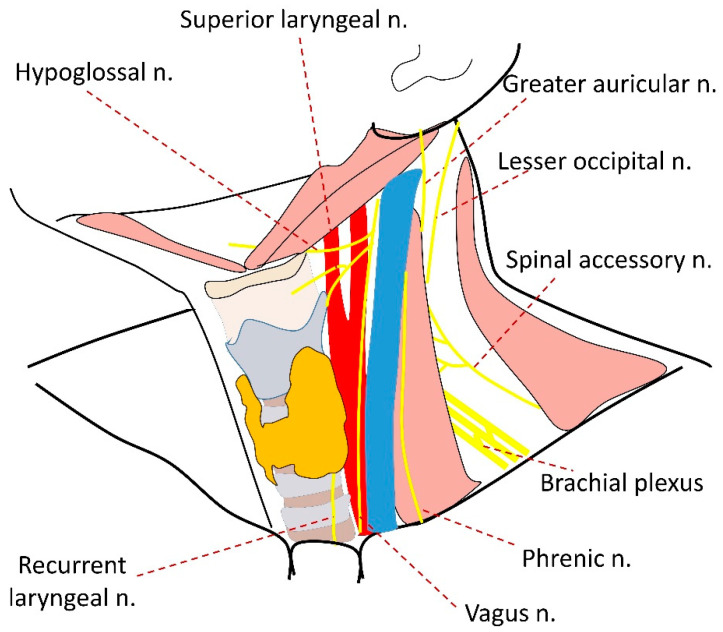
Nerves of the neck region that can potentially develop into schwannomas in proximity to the thyroid gland.

**Table 2 diagnostics-14-02332-t002:** Comparison of imaging technologies in schwannoma diagnostics.

Modality	Disease	Description	Reference
USG	Schwannoma	Hypoechoic mass, regular margins, contextual anechoic areola, multiseptate hypoechoic lesion, macro- and microcalcifications. Contrast-enhanced ultrasonography may be another possibility to distinguish schwannomas; however, data are limited to several case reports. The tumour usually appears as strong, inhomogeneous enhancement both in early and in late phases. Elastography usually indicates dense stiff tissue.	[3,9,10,12,13,25,27,29,31]
	Thyroid cancer	Hypoechoic mass, ill-defined margin, irregular shape, heterogeneity, absence of cystic lesion and/or halo sign, presence of calcification and invasion to other anatomical structures.	
CT	Schwannoma	Mass that can extend to intervertebral foramen. Contrast enhancement may be helpful as thyroid gland tends to accumulate iodine contrast and may reveal separate plane between gland and tumour.	
	Thyroid cancer	Mass with ill-defined borders, extra-thyroid extension, lymph node involvement, or invasion of surrounding structures.	
MRI	Schwannoma	Solid low-intensified and high-intensified tumour by T1- and T2-weighted images, respectively; can be connected to the cervical spinal cord. MRI can delineate mass as distinct from less hyperintense thyroid and enhanced T1-weighted images and demonstrate plane of separation between lesion and thyroid gland.	
	Thyroid cancer	Malignant nodules usually have lower apparent diffusion coefficient value and lower intensity ration on T2-weighted imaging. Similarly, invasion to adjacent structures is indication of malignancy.	
Sci	Schwannoma	Cold nodule in nature or normal thyroid scan.	
	Thyroid cancer	Cold nodule in nature or normal thyroid scan.	

## Data Availability

Data is available on personal request.

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
