# Peer review of "Neck Schwannoma Masking as Thyroid Tumour: Into the Deep of Diagnostics and Anatomy"

_diagnostics, 2024, doi:10.3390/diagnostics14202332_

Round 1

Reviewer 1 Report

Comments and Suggestions for Authors

I would like to congratulate the authors of this case report. 

The differential diagnosis process for schwannomas is not yet based on a standard guideline. It would be advantageous to increase the number of case reports in the literature to facilitate this process.  

Minor comments:

In cases where schwannomas are situated close to the thyroid, they may be mistaken for thyroid tumors. 

Restricting the general information presented in the introduction to the thyroid is unnecessary.

Additionally, schwannomas may be mistaken for parathyroid; the relationship between parathyroid adenoma has already been checked in the case presentation. PTH and calcium levels were evaluated by controlling these variables. 

Even esophageal adenoma, such references as 

* https://doi.org/10.1016/j.circen.2015.10.007

** https://doi.org/10.3389/fendo.2024.1258233

To enhance the general information content and increase the number of citations the article may receive, it would be beneficial to include in the introduction a brief mention of the fact that schwannomas can be confused with other tumors in the thyroid neighborhood. 

Information such as the patient's age and gender should be included. It can also be added by referring to this study in Table 1 (it will be enough to write “this study” as a reference).  

I suggest combining Figures 4 and 5.

In the discussion section, I recommend adding a reference to the paragraph that states, "A complete excision is strongly advised to prevent both the progression of..."

A comparative analysis of FNA and CNB, including examining their respective advantages and disadvantages, proved an effective means of facilitating discussion. The text provides sufficient information regarding the limitations involved. 

The conclusion section, "The benefit of a CNb…" should be corrected.  

It should be noted that histology images do not indicate the degree of magnification. It would be beneficial to include in the figure description the number of images provided with a specified magnification ratio. 

No commentary is provided regarding the histology result. Additionally, supplementary information regarding the WHO grading system should be included. As the diagnosis is based on histology with CNB, the details of the histological examination are not included in the text. To facilitate interpretation of the figure, the legend should include information such as the type of cells (spindle-shaped cells), the shape of the cell nuclei (band-like nuclei), and the magnification. 

Author Response

Schwannomas may be mistaken for parathyroid; the relationship between parathyroid adenoma has already been checked in the case presentation. PTH and calcium levels were evaluated by controlling these variables. 

Even esophageal adenoma, such references as 

* https://doi.org/10.1016/j.circen.2015.10.007

** https://doi.org/10.3389/fendo.2024.1258233

To enhance the general information content and increase the number of citations the article may receive, it would be beneficial to include in the introduction a brief mention of the fact that schwannomas can be confused with other tumors in the thyroid neighborhood. 

-Thank you for the comment, we suplimented the information in the abstract and in the text abouth parathyroid tumors and included the abowementioned references

Information such as the patient's age and gender should be included. It can also be added by referring to this study in Table 1 (it will be enough to write “this study” as a reference).

-Thank you for the comment the information was added to the manuscript

In the discussion section, I recommend adding a reference to the paragraph that states, "A complete excision is strongly advised to prevent both the progression of..."

Thank you, we added this sentence in discussions

I suggest combining Figures 4 and 5.

-Unfortunately we could not do this as the histology figures deteriorate in their size

The conclusion section, "The benefit of a CNb…" should be corrected. 

-Corrected

It should be noted that histology images do not indicate the degree of magnification. It would be beneficial to include in the figure description the number of images provided with a specified magnification ratio. 

-Thank you, we added the corrections in the text

No commentary is provided regarding the histology result. Additionally, supplementary information regarding the WHO grading system should be included. As the diagnosis is based on histology with CNB, the details of the histological examination are not included in the text. To facilitate interpretation of the figure, the legend should include information such as the type of cells (spindle-shaped cells), the shape of the cell nuclei (band-like nuclei), and the magnification. 

-Thank you, we included WHO grading system and suplemental data.

The authors of the manuscript sincerly thank the editor and reviewer for time and efforts to improve the quality of the manuscript.

Reviewer 2 Report

Comments and Suggestions for Authors

This case report describes a patient presenting with a thyroid nodule which was ultimately diagnosed as schwannoma with core needle biopsy as FNA was inconclusive. Excellent imaging pictures are included with the submission.

1.Abstract: Can be focused and improved. Incidence of schwannomas in the central nervous system is not relevant to this case report.  A focus on schwannomas in the neck, especially anterior triangle is necessary.

2.Introduction: Can be focused and improved. Incidence of schwannomas in the central nervous system is not relevant to this case report. Similarly, sentences 44 to 52 can be summarized into a few sentences.

3.Case description: Please mention history and examination findings. Were any thyroid/parathyroid lab tests performed and what were the results? 

4.Discussion: Please correlate discussion with your case report. Did your patient develop "first bite syndrome"? Is it relevant to the current case?

Sentence 187 states that the imaging of neck tumors associated with thyroid cancer....and refers to comparative analysis of Schwannomas. Instead please compare imaging findings in thyroid nodule/cancer vs thyroid schwannoma.  

Comments on the Quality of English Language

Spell check and manuscript editing required.

For example, sentence 217 "neoplasia that patient undergone several FNA with..."

Author Response

1.Abstract: Can be focused and improved. Incidence of schwannomas in the central nervous system is not relevant to this case report.  A focus on schwannomas in the neck, especially anterior triangle is necessary.

-Thank you. we have made several corrections. However, we should note that there is no exact epidemiologicla data about neck schwannomas as they are rare. therefore, we made accent on the available data with some new additional corrections

2.Introduction: Can be focused and improved. Incidence of schwannomas in the central nervous system is not relevant to this case report. Similarly, sentences 44 to 52 can be summarized into a few sentences.

-Thank you, the recommendation for the manuscirpt is 3000 words. The introduction is based on a the fact that the available data is limited and only can be ontained from the information about CNS conditions

3.Case description: Please mention history and examination findings. Were any thyroid/parathyroid lab tests performed and what were the results? 

-Thank you. We have included the information on hormone status, medical history and evaluation.

4.Discussion: Please correlate discussion with your case report. Did your patient develop "first bite syndrome"? Is it relevant to the current case?

-As the neck area is prominent for complications and we did perform differential diagnosis with parathyroid tumours we included data about this complication and consider it relevant.

Sentence 187 states that the imaging of neck tumors associated with thyroid cancer....and refers to comparative analysis of Schwannomas. Instead please compare imaging findings in thyroid nodule/cancer vs thyroid schwannoma.  

Thank you. we have corrected and added new data to the table.

Round 2

Reviewer 1 Report

Comments and Suggestions for Authors

I recommend accepting the article as it is.